# Fargesin Inhibits EGF-Induced Cell Transformation and Colon Cancer Cell Growth by Suppression of CDK2/Cyclin E Signaling Pathway

**DOI:** 10.3390/ijms22042073

**Published:** 2021-02-19

**Authors:** Ga-Eun Lee, Cheol-Jung Lee, Hyun-Jung An, Han Chang Kang, Hye Suk Lee, Joo Young Lee, Sei-Ryang Oh, Sung-Jun Cho, Dae Joon Kim, Yong-Yeon Cho

**Affiliations:** 1BRL & BK21-4th Team, College of Pharmacy, The Catholic University of Korea, 43, Jibong-ro, Wonmi-gu, Bucheon-si, Gyeonggi-do 14662, Korea; rkeoddl520@naver.com (G.-E.L.); veritas0613@kbsi.re.kr (C.-J.L.); anhyunjung11@gmail.com (H.-J.A.); hckang@catholic.ac.kr (H.C.K.); sianalee@catholic.ac.kr (H.S.L.); joolee@catholic.ac.kr (J.Y.L.); 2Research Center for Materials Analysis, Korea Basic Science Institute, 169-148, Gwahak-ro, Yuseong-gu, Daejeon 34133, Korea; 3Natural Medicine Research Center, Korea Research Institute of Bioscience & Biotechnology, 30 Yeongudanji-ro, Ochang-eup, Cheongwon-gun, ChungBuk 363-883, Korea; seiryang@kribb.re.kr; 4School of Medicine, University of Minnesota, Minneapolis, MN 55455, USA; choxx500@umn.edu; 5Department of Molecular Science, University of Texas Rio Grande Valley, Edinburg Research Education Building, Room 2.112, 1214, W. Schunior St, Edinburg, TX 78541, USA; dae.kim@utrgv.edu

**Keywords:** fargesin, cell proliferation, carcinogenesis, chemoprevention, therapeutic compound

## Abstract

Although the lignan compound fargesin is a major ingredient in Shin-Yi, the roles of fargesin in carcinogenesis and cancer cell growth have not been elucidated. In this study, we observed that fargesin inhibited cell proliferation and transformation by suppression of epidermal growth factor (EGF)-stimulated G_1_/S-phase cell cycle transition in premalignant JB6 Cl41 and HaCaT cells. Unexpectedly, we found that signaling pathway analyses showed different regulation patterns in which fargesin inhibited phosphatidylinositol 3-kinase/AKT signaling without an alteration of or increase in mitogen activated protein kinase (MAPK) in JB6 Cl41 and HaCaT cells, while both signaling pathways were abrogated by fargesin treatment in colon cancer cells. We further found that fargesin-induced colony growth inhibition of colon cancer cells was mediated by suppression of the cyclin dependent kinase 2 (CDK2)/cyclin E signaling axis by upregulation of p21^WAF1/Cip1^, resulting in G_1_-phase cell cycle accumulation in a dose-dependent manner. Simultaneously, the suppression of CDK2/cyclin E and induction of p21^WAF1/Cip1^ were correlated with Rb phosphorylation and c-Myc suppression. Taken together, we conclude that fargesin-mediated c-Myc suppression inhibits EGF-induced cell transformation and colon cancer cell colony growth by the suppression of retinoblastoma (Rb)-E2F and CDK/cyclin signaling pathways, which are mainly regulated by MAPK and PKB signaling pathways.

## 1. Introduction

Oriental medicinal herbs and dietary foods contain many useful bioactive ingredients and have been considered as a natural storage containing novel compounds that may have therapeutic value against human diseases. Some of these compounds such as myricetin, quercetin, epigallocatechin gallate, and flavonoids in dietary foods and tea contain chemopreventive effects such as inhibition of cell proliferation and transformation [1] by inhibition of the extracellular signal-regulated kinases (ERKs)/p90 ribosomal S6 kinases (RSKs) signaling pathway in a non-toxic manner [2]. For the last decade, our laboratory has paid attention to the ingredients of Shin-Yi, the dried flower buds of *Magnolia fargesii* Cheng (Magnoliaceae), which has been used in the treatment of emphysema, nasal congestion, sinusitis, and allergic rhinitis [3]. Tetrahydrofurofuranoid lignans including magnolin, aschantin, epimagnolin, dimethoxyaschantin, dimethylliroresinol, dimethylpinoresinol, and fargesin are major and pharmacologically active ingredients of Shin-Yi [3,4,5,6,7,8,9,10]. Previously, our research demonstrated that magnolin targeted the active pockets of ERK1 and ERK2 by forming hydrogen bonds with Lys168 of ERK1 and Met108 and Lys54 of ERK2, resulting in a 68 nM IC_50_ for ERK1 and a 16.5 nM IC_50_ for ERK2 [11]. Moreover, our research interest was expanded to include identifying the molecular targets of aschantin and epimagnolin [12,13,14]. The results demonstrated that those two compounds targeted the active pocket of the mTOR kinase with a less than 400 nM IC_50_ [11,12,13]. However, other Shin-Yi compounds that might have biological activities in diverse human diseases have not been elucidated.

A recent study revealed that fargesin increases basal glucose uptake by induction of phosphorylation of AKT at Ser473 but does not affect AMPK phosphorylation [15]. Interestingly, transactivation activities of the transcription factors nuclear factor-kappa B (NF-ĸB) and activator protein-1 (AP-1), which are well-known transcription factors that regulate multiple pro-inflammatory genes and carcinogenesis, were suppressed by fargesin in a protein kinase C (PKC)-dependent manner [16]. Although the effects of fargesin on anti-inflammatory and antioxidant properties have been elucidated, the role and anticancer effects of fargesin have not been reported. Since mTOR has essential roles in cell proliferation, cell transformation, protein synthesis, and the nutritional level sensory response [17], mTOR allosteric inhibitors that suppress mTORC1 activity have been considered to be able to potentiate in cancer growth suppression. However, the phosphatidylinositol 3-kinase (PI3K)/AKT/mTOR signaling pathway plays a key role in cancer cell proliferation by regulating cell cycle modulation [18], and the inhibition may be toxic to normal cells. Since natural compounds originating from medicinal herbs have generally been found to have low toxicity [19], identification of non-toxic compounds from medicinal herbs that can target the mTOR kinase is important.

The c-Jun protein is a central component of the mammalian transcription factor AP-1 complex and it has a key role in the regulation of diverse cellular processes including cell proliferation, transformation, and cancer development [20]. Since the AP-1 transcription factor regulates about 60% of tumorigenesis-related gene expression in eukaryotic cells, the signaling pathways that regulate AP-1 transactivation activity are of special interest in cancer therapy. Importantly, phosphorylation of the c-Jun C-terminus at Ser243 by glycogen synthase kinase 3 β (GSK3β) is reported to lead to Fbxw7 binding and degradation of c-Jun [21]. Previously, we demonstrated that aschantin targeting of the active pocket of the mTOR kinase suppresses epidermal growth factor (EGF)-induced phosphorylation of AKT at Ser473, resulting in a reduction in the total c-Jun protein level via ubiquitination-dependent destabilization [12]. The reduction in total c-Jun protein levels by aschantin treatment resulted in inhibition of c-Jun/AP-1 transactivation activity and suppression of cell growth in MIAPaCa-2 pancreas and LNCaP prostate cancers [12]. Thus, signaling pathways that regulate c-Jun/AP-1 transactivation activity and protein stability have been considered to have importance as chemopreventive and therapeutic agents of cancer. However, the effects of fargesin on cancer cell growth have not yet been elucidated.

## 2. Results

### 2.1. Fargesin-Mediated G_1_/S Cell Cycle Transition Inhibition Suppresses EGF-Induced Cell Proliferation and Transformation

Fargesin is a lignan compound and a major ingredient of Shin-Yi, along with magnolin, aschantin, and epimagnolin (Appendix A). Although recent research has indicated that fargesin has an important role in the inhibition of inflammatory responses and glucose metabolism, the roles of fargesin in cell proliferation and cell transformation in carcinogenesis have not been fully elucidated. Since JB6 Cl41 mouse skin epidermal cells and HaCaT human skin keratinocytes have been widely used to analyze the potency of chemoprevention and therapeutic compounds in tumor promoter-induced carcinogenesis [20], both were used to initially examine the inhibitory effects of fargesin on cell proliferation. We observed that fargesin suppressed cell proliferation of JB6 Cl41 and HaCaT cells in a dose-dependent manner (Figure 1a). Both JB6 Cl41 and HaCaT cells showed a 50% inhibitory effect of cell proliferation at approximately 22–23 μM fargesin concentration (Figure 1a). Next, we explored the effect of fargesin on cell cycle phase distribution in normal cell culture conditions. We found that fargesin treatment induced the G_1_-phase cell cycle population in JB6 Cl41 and HaCaT cells in a dose-dependent manner (Figure 1b). In contrast, the S-phase cell cycle population was suppressed by fargesin treatment in both JB6 Cl41 and HaCaT cells in a dose-dependent manner (Figure 1b). Additionally, we did not observe the cytotoxic effects of fargesin at 60 μM by 48 h in JB6 Cl41 cells (Appendix A) as well as fargesin-induced apoptosis in JB6 Cl41 and HaCaT cells by 48 h and 72 h, respectively (Appendix A). The cell cycle distribution of JB6 Cl41 and HaCaT cells in normal cell culture conditions was similar to the previous literature [11,13,22]. To examine whether the G_1_-phase cell cycle accumulation of JB6 Cl41 and HaCaT cells treated by fargesin was mediated by a growth factor-induced G_1_/S cell cycle transition or not, we starved the cells and co-treated them with EGF and fargesin as indicated. We observed that EGF stimulation increased the S-phase cell cycle population and reduced the G_1_-phase cell cycle population in JB6 Cl41 and HaCaT cells (Figure 1c). Notably, the EGF-induced S-phase increase was abolished by fargesin treatment in JB6 Cl41 and HaCaT cells in a dose-dependent manner (Figure 1c). In contrast to the S phase, the decreased G_1_-phase cell cycle population by EGF stimulation was increased by fargesin treatment in JB6 Cl41 and HaCaT cells (Figure 1c). Oscillation of cell cycle phases of the G_1_ and S phases was shown to be in accordance with fargesin’s dose dependence (Figure 1c). Since anchorage-independent colony growth is a hallmark of cell transformation from normal cells to cancer cells [20], we further examined the inhibitory activity of fargesin on anchorage-independent cell transformation induced by tumor promoters, such as EGF [11,20]. As expected, we found that JB6 Cl41 and HaCaT cell colony formation in soft agar was significantly induced by EGF (Figure 1d). We also observed that fargesin suppressed the EGF-induced anchorage-independent colony formation in a dose-dependent manner (Figure 1d). These inhibitory effects of fargesin were similarly observed in both JB6 Cl41 and HaCaT cells. Taken together, these results demonstrate that fargesin contains anti-proliferative and anti-carcinogenic potentials by suppression of the tumor promoter-induced G_1_/S-phase cell cycle transition at the promotion stage.

### 2.2. Effects of Fargesin on the Cell Proliferation and Colony Growth of Colon Cancer Cells

Previously, we demonstrated that magnolin and aschantin, which are major ingredients harboring a similar chemical structure to fargesin and abundantly present in Shin-Yi, suppressed the colony growth of lung and prostate cancer cells by targeting the active pockets of ERK1/2 and the mTOR kinase, respectively [11,12]. Moreover, we found that fargesin suppressed cell proliferation and EGF-induced cell transformation (Figure 1). These results suggest that fargesin might have inhibitory effects on cancer cell proliferation and colony growth. Based on this rationale, we examined the effects of fargesin on the cell proliferation and colony growth of colon cancer cells including HCT116, WiDr, and HCT8 cells. We found that fargesin inhibited cell proliferation of these cells in a dose-dependent manner (Figure 2a). However, despite the dose dependency, HCT116, WiDr, and HCT8 showed different sensitivities for fargesin-mediated IC_50_ values for cell proliferation inhibition, about 35, 38, and 45 μM, respectively (Figure 2a). Notably, fargesin treatment increased the G_1_/G_0_-phase cell cycle population and decreased the S-phase cell cycle population in these colon cancer cells (Figure 2b), as similarly shown in premalignant cells such as JB6 Cl41 and HaCaT cells (Figure 1b,c). Since cancer cells already escape from cell-to-cell contact inhibition, we explored fargesin’s effects on the foci formation ability of cancer cells. We found that fargesin inhibited foci formation of colon cancer cells in a dose-dependent manner (Figure 2c, and Appendix A). The IC_50_ of fargesin in foci formation was about 30 μM in HCT116 and WiDr, whereas the IC_50_ value in HCT8 cells was about 40 μM (Figure 2c). Next, we examined the effects of fargesin on the anchorage-independent colony growth of colon cancer cells. As expected, fargesin suppressed the colony growth of colon cancer cells in a dose-dependent manner (Figure 2d, and Appendix A). Interestingly, WiDr colon cancer cells showed stronger sensitivity for fargesin compared to HCT116 and HCT8 cells. The effects of fargesin on colon cancer cell proliferation and colony growth suggested to us that fargesin contains not only a chemopreventive, but also a therapeutic potential for tumor promoter-induced cell transformation and colon cancer therapy, respectively.

### 2.3. Fargesin Induced Morphological Changes by Suppression of PI3K/PKB Signaling Pathway

Previous studies indicated that fargesin did not alter the morphological change in cells in vitro and in vivo [23,24]. However, we observed that fargesin induced a significant morphological change in JB6 Cl41 and HCT8 colon cancer cells (Figure 3a). The cells became longish and emaciated in JB6 Cl41 cells (Figure 3a, left panels) and flattened and enlarged in HCT8 colon cancer cells (Figure 3a, right panels). Since fargesin attenuated cell proliferation (Figure 2), we hypothesized that the morphological change might be involved in cell proliferation and colony growth. To address this question, we firstly examined the phosphoprotein profiles of signaling molecules, focusing on the ERKs and the PI3K/AKT signaling pathways in JB6 Cl41 cells. We found that phosphorylation of mitogen-activated protein kinase kinase (MEK), an upstream kinase of ERK1/2, was induced by the EGF treatment but was not altered by fargesin treatment (Figure 3b). Interestingly, the unexpected results repeatedly show that EGF-induced ERKs’ and RSKs’ phosphorylation was increased by fargesin treatment in a dose-dependent manner (Figure 3b). In addition, total protein levels of MEKs, ERKs, and RSKs were not changed by EGF alone or EGF/fargesin combined treatment (Figure 3b). By analyzing the PKB signaling pathway, EGF-induced mTOR phosphorylation at Ser2448 and AKT phosphorylation at Thr308 and Ser473 were observed (Figure 3c), as shown in our previous reports [11,12]. In contrast to ERKs and RSKs, phosphorylation of mTOR at Ser2448 and AKT at Thr308 and Ser473 was decreased by fargesin treatment in a dose-dependent manner (Figure 3c). However, we did not know the reason why AKT phosphorylation at Thr308 by co-treatment of EGF and 15 μM of fargesin was increased (Figure 3c). These results suggest that fargesin might target a signaling molecule in the PI3K/protein kinase B (PKB) pathway. To examine whether cancer cells also showed similar phosphorylation patterns for ERKs and PI3K/PKB signaling pathways, or not, we conducted Western blot analysis. We found that fargesin suppressed phosphorylation levels of MEKs, ERKs, and RSKs in WiDr and HCT8 colon cancer cells (Figure 3d). Similarly, phosphorylation levels of PI3K/PKB signaling molecules were inhibited by fargesin treatment in WiDr and HCT8 colon cancer cells (Figure 3e). These results indicate that fargesin-mediated inhibition of ERKs and PKB signaling pathways suppresses colon cancer cell proliferation and colony growth.

### 2.4. Modulation of CDKs/Cyclins-p21^WAF1/Cip1^ Signaling Axis by Fargesin Induces G_1_/G_0_ Cell Cycle Accumulation

Our previous results demonstrated that fargesin inhibited the cell proliferation and colony growth of colon cancer cells by induction of G_1_/G_0_ cell cycle accumulation (Figure 2). The cell proliferation and colony growth inhibitions might be mediated via inhibition of the growth factor-mediated ERKs and PI3K/PKB signaling pathways in colon cancer cells (Figure 3). We further found that fargesin dramatically induced the G_1_/G_0_-phase cell cycle population and reduced the S-phase cell cycle population compared to non-treated control cells (Figure 2b). These results indicated that the fargesin-mediated cell proliferation and colony growth inhibitions in colon cancer cells were associated with impairment of the G_1_/S cell cycle progression. ERKs and PKB signaling pathway-mediated CDKs and cyclins generally regulate cell proliferation and survival by the mediation of gene expression and protein synthesis [2,11,12,18]; however, it is not clear whether cell cycle control by fargesin is mediated by the cell cycle regulators such as CDK2/4 and cyclin D/E. Based on this rationale, we analyzed the protein levels of CDK2/4 and cyclin D1/E in colon cancer cells. We observed that CDK4 was reduced in HCT8 and HCT116, but not in WiDr, by fargesin treatment (Figure 4a). Moreover, cyclin D1 was downregulated in WiDr and HCT116, but not in HCT8, by fargesin treatment (Figure 4a). The results suggest that the CDK4 and cyclin D1 signaling pathway may partially involve fargesin-mediated cell cycle regulation. In contrast, CDK2 and cyclin E protein levels were dramatically decreased in WiDr, HCT8, and HCT116 colon cancer cells by fargesin treatment in a dose-dependent manner (Figure 4a). Since p21^WAF1/Cip1^ is a well-known regulator for CDK2/cyclin E signaling-mediated cell cycle regulation, we examined whether fargesin-mediated G_0_/G_1_ accumulation of the cell cycle phase and CDKs/cyclins downregulation are related to p21^WAF1/Cip1^ protein levels or not. We found that fargesin induced p21^WAF1/Cip1^ protein levels in WiDr, HCT8, and HCT116 colon cancer cells in a dose-dependent manner (Figure 4b). Similarly, we also found that fargesin induced p21^WAF1/Cip1^ protein levels in JB6 Cl41 cells in a dose-dependent manner (Figure 4c). These results were supported by our immunocytofluorescence assays indicating that p21^WAF1/Cip1^ protein levels were highly detected in the cytoplasm as well as the nucleus with fargesin treatment in WiDr and HCT8 colon cancer cells in a dose-dependent manner (Figure 4d). Taken together, these results suggest that fargesin induced complex formation of p21^WAF1/Cip1^/CDK2/cyclin E, resulting in impairment of the G_1_/S cell cycle transition and inhibition of cell proliferation of premalignant JB6 Cl41 and HaCaT cells, as well as WiDr, HCT8, and HCT116 colon cancer cells.

### 2.5. Differential Regulation of AP-1 Signaling and C-Myc-Mediated CDK2/Cyclin E Signaling Mediates Cell Proliferation Differentially in Premalignant and Cancer Cells

Growth factor-stimulated cell cycle progression for the G_1_/S transition plays a key role in normal controlled cell proliferation and abnormal cancerous cell transformation. Based on the previous literature, transactivation activities of AP-1, a dimer of Fos and Jun, and c-Myc play a pivotal role in G_1_/S cell cycle transition. Since our research group has published that tumor promoters such as EGF or TPA induce AP-1 transactivation activity-mediated cell proliferation and transformation in JB6 Cl41 cells, we identified c-Fos and c-Jun protein levels to be important targets for signaling molecules during EGF-induced cell proliferation and transformation. Moreover, a large body of studies has also shown that c-Myc can modulate the expression of genes controlling the cell cycle, especially the G_1_-phase CDKs, and primarily CDK2, and thereby facilitate transit through the G_1_/S transition [25]. Thus, we analyzed c-Jun and c-Fos protein levels in JB6 Cl41 cells. We found that EGF stimulation induced c-Jun and c-Fos levels in JB6 Cl41 cells (Figure 5a). Notably, JB6 Cl41 cells showed a decrease in c-Jun and c-Fos protein levels by fargesin treatment in a dose-dependent manner, whereas HaCaT cells showed a decrease in only c-Fos, but not in c-Jun, protein levels by fargesin in a dose-dependent manner (Figure 5a). Notably, by the AP-1 luciferase reporter assay, we confirmed that fargesin also suppressed AP-1 transactivation activity in these cells in a dose-dependent manner (Figure 5b). Interestingly, protein level changes of c-Jun and c-Fos were not observed after fargesin treatment of HCT8 and WiDr colon cancer cells in normal cell culture conditions (Figure 5c) as well as EGF-stimulated conditions (Figure 5d). Meanwhile, c-Myc protein levels were dramatically reduced by fargesin treatment in HCT8 and WiDr colon cancer cells in a dose-dependent manner (Figure 5e). Notably, phosphorylated retinoblastoma (Rb) protein levels were inhibited by fargesin treatment in a dose-dependent manner, while total levels of the Rb protein in steady state were maintained (Figure 5f). Since the Myc promoter contains an E2F binding site [26] and fargesin suppresses G_1_/S cell cycle transition in colon cancer cells, we examined the c-Myc protein levels at the nucleus and cytosol by immunocytofluorescence (Appendix A). The results indicate that the c-Myc protein mainly observed in the nucleus in the non-treated control group had a weak intensity (Figure 5g, left panels, and Appendix A) and showed diffusion into the cytosol (Figure 5g, bottom panels, and Appendix A). The normalized intensity of green fluorescence showed that c-Myc protein levels were dramatically reduced by fargesin treatment (Figure 5g, graphs). Taken together, our results demonstrate that fargesin suppressed EGF-induced cell proliferation and cell transformation in premalignant cells and cell proliferation and colony growth of colon cancer cells by modulation of cell cycle regulators via CDK2/cyclin E/p21^WAF1/Cip1^ complex formation. Thus, fargesin can potentiate for chemopreventive and/or therapeutic application.

## 3. Discussion

Within its major ingredients, Shin-Yi extract contains eight different lignan compounds including eudesmin, magnolin, lirioresinol, epimagnolin, aschantin, kubusin, fargesin, and burchellin, which have very similar chemical structures. Shin-Yi has been used as an oriental medicine, generally in order to treat inflammation-mediated diseases including nasal congestion associated with headache, sinusitis, and allergic rhinitis [27]. Although the effects of a medicinal cocktail that includes Shin-Yi are well-known, it is unclear which chemical compounds produce the medicinal effects. Moreover, the molecular targets and action mechanisms have not been clearly described. Our laboratory has undertaken studies to identify the molecular targets of some active natural compounds originating from oriental medicinal herbs, which have revealed the molecular action mechanisms through combinational approaches of molecular biology and biochemistry along with computational and structural prediction analysis. Previously, we reported that magnolin targets the active pocket of ERK1 and ERK2 with about 68 and 16.5 nM in IC_50_ values, respectively [11]. The molecular biology and biochemical analyses demonstrated that magnolin competed with ATP at the active pocket of ERK2 [11], whereas the computational and structural prediction analyses suggested that magnolin formed hydrogen bonds with Lys 168 of ERK1 and Met168 and Lys54 of ERK2 [11]. Moreover, aschantin, a major ingredient of Shin-Yi with a similar chemical structure to that of magnolin, targets the active pocket of the mTOR kinase [12]. Aschantin was shown to form a hydrogen bond with Val2240 of the mTOR kinase domain [12]. Interestingly, magnolin did not alter the phosphoprotein profiles of MAPK signaling molecules [12]. Moreover, aschantin did not inhibit protein kinase activities of PI3K, PDK-1, or AKT [11,12]. Recently, our research group reported that epimagnolin, a stereochemical epimer of magnolin, targets the active pocket of the mTOR kinase, but not that of ERK1 and ERK2, contrasting with our expectation that epimagnolin could inhibit the activity of ERK1 and ERK2 [13]. These results demonstrated that slight differences in chemical structure might produce large differences in the molecular targeting of proteins that are involved in human diseases including cancers. Although molecular targeting of fargesin was not conducted in this study, signaling studies using fargesin have demonstrated that fargesin stimulates glucose uptake and consumption by GLUT4 translocation to the cytoplasmic membrane [15]. Moreover, fargesin inhibits pro-inflammatory modulators such as cyclooxygenase-2 and inducible nitric oxide synthase, resulting in suppression of inflammatory responses in THP-1 monocytes. Notably, production of interleukin-1β and tumor necrosis factor-β, pro-inflammation cytokines, and CCL-5, a chemokine, was inhibited by fargesin [16]. Importantly, transactivation activities of the transcription factors nuclear factor-kappa B (NF-ĸB) and activator protein-1 (AP-1), which are well-known transcription factors that regulate multiple pro-inflammatory genes and carcinogenesis, were suppressed by fargesin in a protein kinase C (PKC)-dependent manner [16]. Our results demonstrate that fargesin suppressed AP-1 transactivation activity (Figure 5a,b), resulting in EGF-induced cell transformation (Figure 1d). Since fargesin suppressed AKT phosphorylation at Thr308 and Ser473, we suggest that the target molecule(s) of fargesin may be an AKT upstream, including receptor tyrosine kinases at the cytoplasmic membrane.

The *c-fos* and *c-jun* proto-oncogenes, components of the AP-1 transcription factor, are essential transcription factors involved in cell transformation and tumorigenesis [28,29]. CDK2, a serine/threonine kinase, has a role in the G_1_/S-phase cell cycle progression, DNA synthesis initiation, and the exit regulation from the S phase. In the G_1_/S-phase transition process, the CDK4/6-cyclin D complex initially phosphorylates the Rb protein [30,31]. After this event, CDK2 and cyclin E associate and the CDK2/cyclin E complex phosphorylates Rb [32], resulting in the release of E2F transcription factors to prepare for gene expression in the G_1_/S-phase cell cycle transition and in DNA synthesis [23]. Thus, since the CDK2/cyclin E complex is maximal in the G_1_ cell cycle phase, CDK2 and cyclin E protein levels are critically important in the prediction of cell cycle progression. In the present study, our results show that fargesin treatment reduced total protein levels of CDK2 and cyclin E (Figure 4a), which are critical components of the G_1_/S-phase cell cycle transition [18]. Other recent results have indicated that Jun D knockdown can reduce the protein levels of cell cycle regulators including c-Myc, CDK4, and CDK2 in prostate cancer cells [24]. In contrast, the protein levels of c-Myc, CDK4, and CDK2 were increased by overexpression of Jun D in prostate cancer cells [24]. These results suggest that AP-1-mediated c-Myc and CDKs have key roles in the G_1_/S-phase cell cycle transition. Surprisingly, in this study, we observed that fargesin treatment reduced the protein levels of c-Fos and c-Jun (Figure 5a), resulting in the suppression of AP-1 transactivation activity (Figure 5b), which was measured by firefly luciferase activity using cell lysates from stable cells harboring the AP-1 luciferase report plasmid in the genome [14]. Notably, the total protein levels of c-Myc and CDK2 were reduced by fargesin treatment in colon cancer cells (Figure 5e,g and Figure 4a). These results demonstrate that fargesin-mediated cell cycle modulation inhibits cell proliferation via AP-1-mediated gene expression of cell cycle regulators such as c-Myc and CDK2.

Interestingly, we found that fargesin treatment did not reduce the phosphorylation levels of signaling molecules in MAPK. In contrast to our hypothesis, fargesin increased the phosphorylation levels of ERK1/2 and RSKs (Figure 3b). Furthermore, a previous publication indicated that fargesin induces AKT phosphorylation, and wortmannin, a PI3K inhibitor, showed conversely suppressed fargesin-induced AKT phosphorylation [14]. However, we showed that fargesin inhibited EGF-induced phosphorylation levels of mTOR and AKT in JB6 Cl41 premalignant (Figure 3c) and WiDr and HCT8 colon cancer cells (Figure 3e). Importantly, we further found that fargesin increased the total protein level of p21^WAF1/Cip1^ (Figure 4b,c), but not of p16^INK4a^ (data not shown), suggesting that fargesin target signaling is via a CDK2/cyclin E rather than a CDK4/cyclin D signaling cascade (Figure 4a). This suggestion is supported by recent results finding that c-Myc induces cyclin E and suppresses p21^WAF1/Cip1^ [24], resulting in the induction of cell proliferation. Moreover, gene expression aberration of CDK2 and cyclin E affects the G_1_/S-phase cell cycle progression [18]. Our results demonstrate that fargesin inhibits cell proliferation via suppression of the G_1_/S-phase cell cycle transition (Figure 1b,c and Figure 2b), resulting in abrogation of EGF-induced cell transformation. We also observed G_1_-phase cell accumulation following fargesin treatment in HCT8, WiDr, and HCT116 colon cancer cells. Notably, fargesin suppressed the colony growth of colon cancer cells in an anchorage-independent condition. Unfortunately, the involvement of p53 phosphorylation at Ser37 as an upstream transcription factor of p21^WAF1/Cip1^ has not been clearly elucidated. Taken together, the results of this study strongly indicate that fargesin has potential as a chemoprevention and/or therapeutic agent against human cancer.

## 4. Materials and Methods

### 4.1. Chemicals and Antibodies

Chemicals utilized for the molecular and cellular biology investigations and for buffer preparation were purchased from Sigma-Aldrich (St. Louis, MO, USA). Cell culture media including RPMI 1640, modified Eagle’s medium, and Dulbecco’s modified Eagle’s medium were purchased from Corning (Corning, New York, NY, USA). Fetal bovine serum (FBS) and supplements including penicillin and streptomycin were purchased from Life Science Technologies (Rockville, MD, USA) and were heat inactivated before utilization. Dimethyl sulfoxide (DMSO, Sigma-Aldrich) was purchased from Sigma-Aldrich (St. Louis, MO, USA). Recombinant EGF was purchased from ThermoFisher Scientific Korea (Gangnam, Seoul, Korea). Antibodies for phosphoproteins and total proteins in the MAPK signaling pathway including MEKs, ERK1/2, and RSKs were purchased from Cell Signaling Technology (Beverly, MA, USA). Antibodies for PKB signaling and cell cycle regulation pathways including phospho-AKT (S473), phospho-AKT (T308), total-CDK4, and total-cyclin D1 were purchased from Cell Signaling Technology (Beverly, MA, USA) to use in Western blot analyses. Phospho- and total-antibodies for transcription factors including phospho-c-Jun (S63), phospho-c-Jun (S73), and total-c-Jun were purchased from Cell Signaling Technology (Beverly, MA, USA). Antibodies including c-Myc, p21^WAF1/Cip1^, p16^INK4a^, CDK2, cyclin E, and β-actin were from Santa Cruz Biotechnology (Santa Cruz, CA, USA).

### 4.2. Fargesin

Fargesin, with a >99.0% product purity confirmed by performing high-performance liquid chromatography, was generously provided by Dr. SR Oh of the Korean Research Institute of Bioscience and Biotechnology (Korea patent # 10-0321212-0000) [5] as described previously. The fargesin stock solution (100 mM: 1000×) was prepared by dissolving it in DMSO, after which aliquots were prepared and then stored at −20 °C. Fargesin working solutions were freshly diluted in DMSO before utilization. When fargesin was treated into cells, cells were treated upon medium exchange with a premixed fargesin cell culture medium. The DMSO concentration did not exceed 0.1% of the total final volume.

### 4.3. Cell Culture

All cells including premalignant JB6 Cl41 and HaCaT cells and HCT8, WiDr, and HCT116 colon cancer cells were obtained from the American Type Culture Collection (Manassas, VA, USA). The JB6 Cl41 cells were cultured in MEM supplemented with final 5% FBS. The HCT116 and HaCaT cells were cultured in DMEM supplemented with final 10% FBS. The colon cancer cells, including HCT8 and WiDr, were cultured in RPMI-1640 supplemented with final 10% FBS. Cell lines were periodically authenticated by monitoring cell morphology, by growth curve analysis, and by undertaking mycoplasma contamination inspection. When cells reached 90% confluence, the cells were split for passage prior to seeding for experiments. Fresh complete medium was refed every other day.

### 4.4. Cell Proliferation Assay

To measure cell proliferation, JB6 Cl41 cells (1 × 10^3^), HaCaT cells (0.65 × 10^3^), HCT8 cells (1 × 10^3^), WiDr cells (1 × 10^3^), or HCT116 cells (2 × 10^3^) were seeded into 96-well plates (100 µL of cell culture medium) and cultured overnight, and absorbance was measured (0 h) at 492 nm followed by the 3-(4,5-dimethylthiazol-2-yl)-5-(3-carboxymethoxyphenyl)-2-(4-sulfophenyl)-2H-tetrazolium (MTS)-based CellTiter 96^®^ Aqueous One Solution assay according to the manufacturer’s suggested protocols (Promega, Madison, WI, USA). Briefly, 20 µL of the MTS solution was added to each well and incubated for 1 h at 37 °C in a 5% CO_2_ incubator, and then the reaction was stopped by adding 25 µL of 10% sodium dodecyl sulfate (SDS) solution to each well. The absorbance was measured immediately at 492 nm using an xMark microplate spectrophotometer (Bio-Rad Laboratories, Hercules, CA, USA). The effects of fargesin on cell proliferation were evaluated by comparing the absorbance of the samples to a DMSO-treated control group at 24-h intervals.

### 4.5. Cytotoxicity Assay

JB6 Cl41 cells (1 × 10^4^) were seeded into 96-well plates and cultured overnight, and cell viability was measured at 0 h by an MTS assay as described in “Cell Proliferation Assay”. The cells were treated with fargesin by exchanging medium containing the indicated doses of fargesin. The cell viability was measured at 24 and 48 h. The absorbance was measured immediately at 492 nm using an xMark microplate spectrophotometer (Bio-Rad Laboratories). The effects of fargesin on cell viability were evaluated by comparing the absorbance of the samples to a DMSO-treated control group.

### 4.6. Cell Cycle Analysis

JB6 Cl41 cells (5 × 10^5^), HaCaT cells (2.5 × 10^5^), HCT8 cells (2 × 10^5^), WiDr cells (2 × 10^5^), or HCT116 (2 × 10^5^) cells were seeded into 60-mm dishes and cultured overnight. To analyze the effect of fargesin on the cell cycle distribution under normal cell culture conditions, JB6 Cl41 and HaCaT cells were treated with fargesin in complete cell culture medium for 24 h. The effect of fargesin on the cell cycle phase transition by EGF was examined by exchange of starvation medium (MEM supplemented with 0.1% FBS) overnight following treatment of EGF or co-treatment of EGF and indicated doses of fargesin for 24 h. The cells were trypsinized, fixed, treated with RNase A (100 μg/mL), and then incubated at room temperature for 1 h. The population in each cell cycle phase was measured by propidium iodide (20 μg/mL) staining and flow cytometry followed flow cytometry (BD FACSCalibur™ flow cytometer, Franklin Lakes, NJ, USA).

### 4.7. Western Blotting

Cell lysates were extracted by freezing and thawing in cell lysis buffer (50 mM Tris pH 8.0, 150 mM NaCl, 1% NP-40, Roche protease inhibitor cocktail). Equal amounts of protein (generally 20–30 μg/lane) were resolved by SDS-PAGE, transferred onto polyvinylidene difluoride (PVDF, Merck Millipore Ltd., Burlington, MA, USA) membranes, blocked in a blocking buffer (5% skim milk/1× PBS), and then hybridized with specific antibodies. Blots were washed three times in 1× TBS-T buffer and rehybridized with the appropriate HRP-conjugated secondary antibodies. The proteins were visualized with an enhanced chemiluminescence detection system (Amersham Biosciences, Piscataway, NJ, USA).

### 4.8. Luciferase Assay

For the AP-1 luciferase assay, JB6 Cl41 cells (5 × 10^4^ cells) stably transfected with AP-1 luciferase reporter plasmids were cultured in 24-well plates. The cells were treated with the indicated doses of fargesin for 12 h. The firefly luciferase activity was measured by luminescence detection system (VICTOR X3 plate reader, PerkinElmer, Waltham, MA, USA) using equal volumes of the cell lysates.

### 4.9. Anchorage-Independent Cell Transformation Assay

JB6 Cl41 and HaCaT cells were utilized to evaluate the inhibitory effects of fargesin on EGF-induced cell transformation. The cells (8 × 10^3^ cells/well/6-well plates) were exposed to EGF alone (10 ng/mL) or equal concentrations of EGF and the indicated doses of fargesin in 1 mL of 0.3% top agar composed of basal medium Eagle agar containing 10% FBS. The cultures were continuously cultured for 10–14 days. The cell colonies were observed under an ECLIPSE Ti inverted microscope and scored using the NIS-Elements AR (V. 4.0) computer software program (NIKON Instruments Korea, Gangnam, Seoul, Korea).

### 4.10. Colony Growth Assay of Cancer Cells (Soft Agar Assay)

The inhibitory effects on cancer colony growth by fargesin treatment were assessed using HCT8, WiDr, and HCT116 colon cancer cells. Briefly, colon cancer cells (8 × 10^3^) suspended in appropriate complete culture medium supplemented with 10% FBS were added to 0.3% top agar containing the indicated doses of fargesin into 6-well plates. The cultures were continuously cultured for 10–14 days. The cell colonies were observed under an ECLIPSE Ti inverted microscope and scored using the NIS-Elements AR (V. 4.0) computer software program (NIKON Instruments Korea).

### 4.11. Focus Formation Assay

HCT8 cells (1 × 10^3^), WiDr cells (1 × 10^3^), or HCT116 cells (1 × 10^3^) were seeded into 6-well plates and cultured overnight. The cells were continuously maintained with complete medium supplemented with the indicated doses of fargesin for 7–10 days. The cells were subjected to Crystal Violet staining (0.05% *w/v* Crystal Violet, 1% formaldehyde, 1× PBS, 1% methanol) and destained. The formed foci were scanned by using an EPSON scanner and were scored by using Image J software.

### 4.12. Immunocytofluorescence Assay

HCT8 cells (4 × 10^4^) were seeded into four-chamber culture slides and cultured for 24 h. Subsequently, the cells were starved for 18 h, pretreated with the indicated doses of fargesin for 30 min, and co-treated with EGF and fargesin for 30 min as indicted. The cells were fixed, permeabilized, and hybridized with the specific antibody overnight at 4 °C in a humidified chamber. The proteins were visualized by using secondary antibody hybridization conjugated with Alexa 488 or 568 under an LSM 710 laser scanning confocal microscope (Carl Zeiss, Oberkochen, Germany).

## 5. Conclusions

The physiological and preventive roles of fargesin in colon cancer cells were clearly shown by cancer cell proliferation and colony growth assays. Moreover, fargesin-mediated inhibition of colon cancer cell proliferation and colony growth is positively co-related to G_1_/S cell cycle transition inhibition. The molecular mechanism studies suggest that CDK2/cyclin E and CDK4/cyclin D are involved in the inhibition of colon cancer cell proliferation by fargesin via p21^WAF1/Cip1^ and c-Myc, respectively.

## Figures and Tables

**Figure 1 ijms-22-02073-f001:**
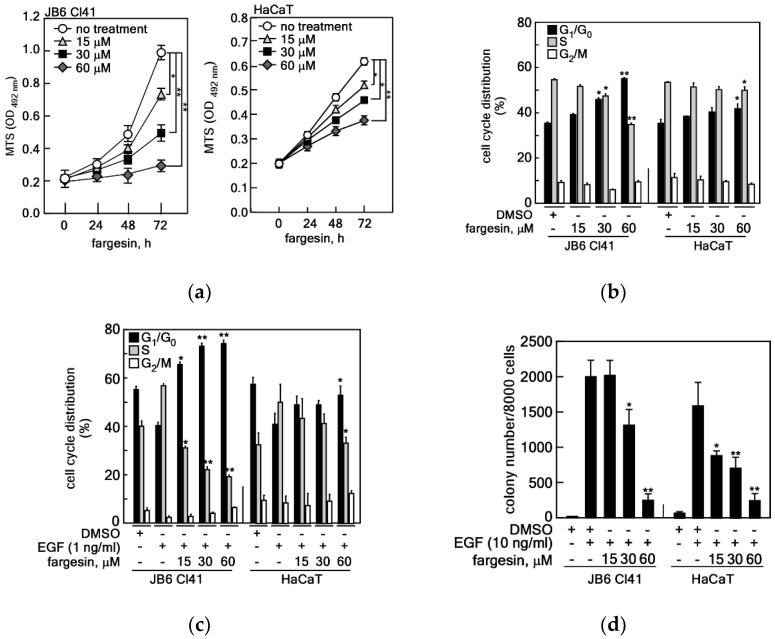
Fargesin-mediated G_1_/S cell cycle transition inhibition suppresses EGF-induced cell proliferation and transformation. (**a**) Inhibitory effects of fargesin on cell proliferation of JB6 Cl41 and HaCaT cells. JB6 Cl41 (1 × 10^3^ cells) and HaCaT (0.65 × 10^3^ cells) were seeded into 96-well cell culture plates, cultured overnight, and treated with indicated doses of fargesin. Cell proliferation was measured at indicated times by MTS assay as described in “Material and Methods”. (**b**) The effects of fargesin on the cell cycle phase distribution of JB6 Cl41 and HaCaT cells. JB6 Cl41 (5 × 10^5^ cells) and HaCaT (2.5 × 10^5^ cells) were seeded into 60-mm cell culture dishes, cultured overnight, and treated with indicated doses of fargesin for 24 h. The cell cycle phase population was measured by flow cytometry. (**c**) The effects of fargesin on cell cycle transition induced by EGF stimulation in JB6 Cl41 and HaCaT cells. JB6 Cl41 (5 × 10^5^ cells) and HaCaT (2.5 × 10^5^ cells) were seeded into 60-mm cell culture dishes, cultured overnight, starved for 24 h by feeding 0.1% FBS, and treated with indicated doses of EGF and fargesin for 24 h. The cell cycle phase population was measured by flow cytometry. (**d**) The effects of fargesin on the EGF-induced cell transformation in JB6 Cl41 and HaCaT cells. JB6 Cl41 (8 × 10^3^ cells) and HaCaT (8 × 10^3^ cells) were mixed with 0.32% top agar supplemented with indicated doses of EGF and fargesin and poured onto bottom agar, which contained indicated doses of EGF and fargesin. After the top agar hardened, the cells were cultured for 10–14 days at 37 °C, in a 5% CO_2_ incubator. The colonies were observed under an ECLIPSE Ti inverted microscope and scored using the NIS-Elements AR (V. 4.0) computer program. (**a**–**d**) Data: a triplicate experiment; values: ±SEM; significance: * *p* < 0.05, ** *p* < 0.01 vs. non-treated control by Student’s *t*-test.

**Figure 2 ijms-22-02073-f002:**
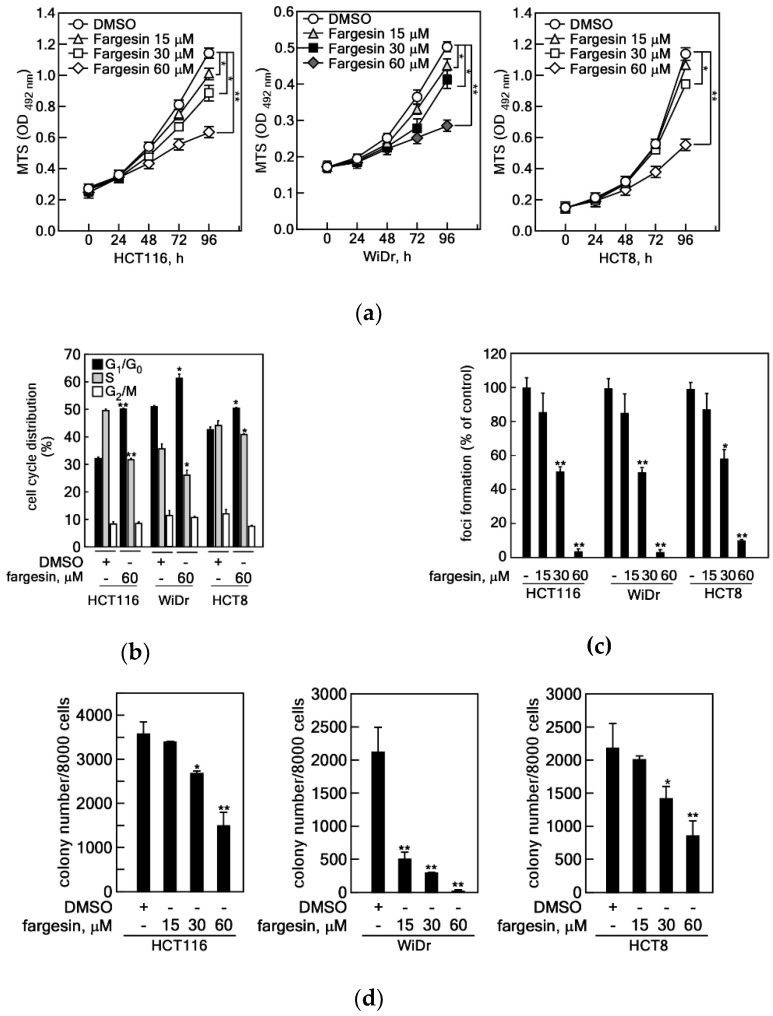
Effects of fargesin on the cell proliferation and colony growth of colon cancer cells. (**a**) Fargesin inhibits cell proliferation of colon cancer cells. HCT116 (2 × 10^3^), WiDr (1 × 10^3^), and HCT8 (1 × 10^3^) colon cancer cells were seeded into 96-well plates, cultured overnight, and treated with indicated doses of fargesin. Cell proliferation inhibition was measured by MTS assay with 24-h intervals up to 96 h. (**b**) Fargesin induces the G_1_ phase of and suppresses the S phase of the cell cycle population in colon cancer cells. HCT116 (2 × 10^5^), WiDr (2 × 10^5^), and HCT8 (2 × 10^5^) colon cancer cells were seeded into 60-mm cell culture dishes, cultured overnight, and treated with fargesin (60 µM) for 24 h. The cell cycle phase population was measured by flow cytometry. (**c**) The effects of fargesin on the foci formation of colon cancer cells. HCT8 (1 × 10^3^), WiDr (1 × 10^3^), and HCT116 (1 × 10^3^) cells were seeded into 6-well cell culture plates, cultured overnight, and treated with indicated doses of fargesin. The cells were cultured for 7–10 days and foci formation was visualized by crystal violet staining/destaining. (**d**) The effects of fargesin on the colony growth of colon cancer cells. HCT116 (8 × 10^3^), WiDr (8 × 10^3^), and HCT8 (8 × 10^3^) colon cancer cells were mixed with 0.32% top agar supplemented with indicated doses of fargesin and poured onto bottom agar, which contained indicated doses of fargesin. After the top agar hardened, the cells were cultured for 10–14 days at 37 °C, in a 5% CO_2_ incubator. The colonies were observed under an ECLIPSE Ti inverted microscope and scored using the NIS-Elements AR (V. 4.0) computer program. (**a**–**d**) Data: a triplicate experiment; values: ±SEM; significance: * *p* < 0.05, ** *p* < 0.01 vs. non-treated control by Student’s *t*-test.

**Figure 3 ijms-22-02073-f003:**
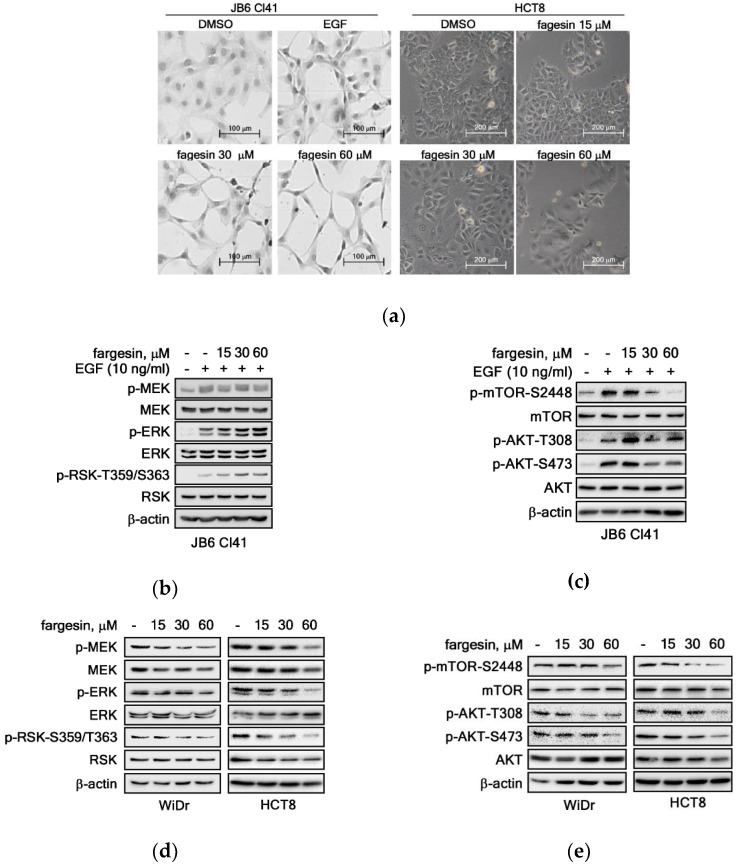
Fargesin induced morphological changes by suppression of PI3K/PKB signaling pathway. (**a**) Fargesin induces morphology changes in cells. JB6 Cl41 (5 × 10^4^) and HCT8 (5 × 10^4^) cells were seeded into 6-well plates, cultured overnight, and treated with indicated doses of fargesin. Representative photographs were obtained by observation under a light microscope (×200). (**b**–**e**) Effects of fargesin on the phosphorylation of ERKs and PI3K/AKT signaling pathways. Phosphoprotein profiles of MEKs/ERKs/RSKs signaling pathway (**b**,**d**) and PI3K/AKT signaling pathway (**c**,**e**) in JB6 Cl41 (**b**,**c**) and WiDr and HCT8 colon cancer cells (**d**,**e**). Total proteins (20 µg) extracted from JB6 Cl41 and colon cancer cells were resolved by SDS-PAGE and transferred onto PVDF membrane. The indicated proteins were visualized by probing the specific antibodies. β-actin was used to verify equal protein loadings.

**Figure 4 ijms-22-02073-f004:**
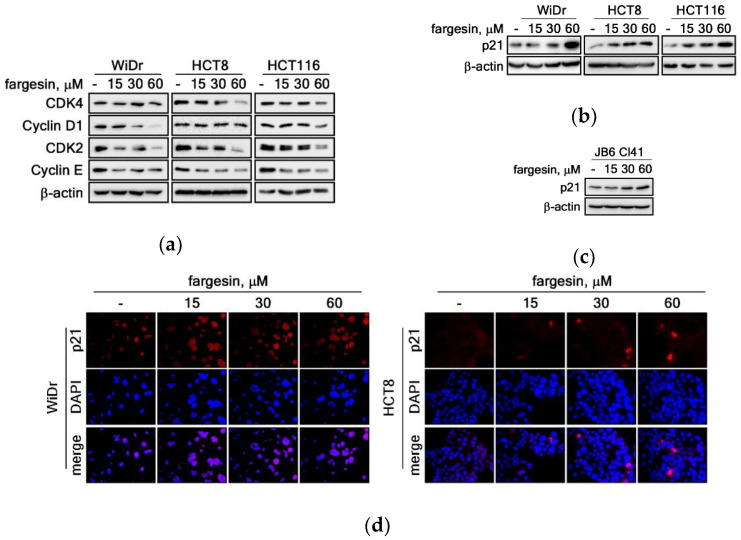
Modulation of CDKs/cyclins-p21^WAF1/Cip1^ signaling axis by fargesin induces G_0_/G_1_ cell cycle accumulation. (**a**) Fargesin suppresses protein levels of cell cycle regulators involved in G_1_/S cell cycle transition. Total proteins (30 μg) extracted from the indicated cells were separated by SDS-PAGE, and the indicated proteins were visualized by probing the specific antibodies indicated. β-actin was used to verify equal protein loadings. (**b**,**c**) Fargesin induces p21^WAF1/Cip1^ protein levels in colon cancer cells. Total proteins (30 μg) extracted from WiDr, HCT8, and HCT116 colon cancer cells (**b**) and JB6 Cl41 cells (**c**) were resolved by SDS-PAGE and visualized by probing the p21^WAF1/Cip1^ antibody and the HRP-conjugated secondary antibody. β-actin was used to verify equal protein loadings. (**d**) Fargesin induces a p21^WAF1/Cip1^-positive cell population. WiDr (1 × 10^4^ cells) and HCT8 (1 × 10^4^ cells) colon cancer cells were seeded into 4-chamber slides, cultured for 48 h, and treated with indicated doses of fargesin. The cells were fixed with 4% paraformaldehyde, permeabilized, and hybridized with a p21^WAF1/Cip1^ specific antibody and Alexa 568-conjugated secondary antibody. The cells were observed under a fluorescence confocal microscope (×200). Representative photographs were obtained from a photograph taken randomly at four different locations/chamber in three independent experiments. Red, p21^WAF1/Cip1^; blue, nuclear.

**Figure 5 ijms-22-02073-f005:**
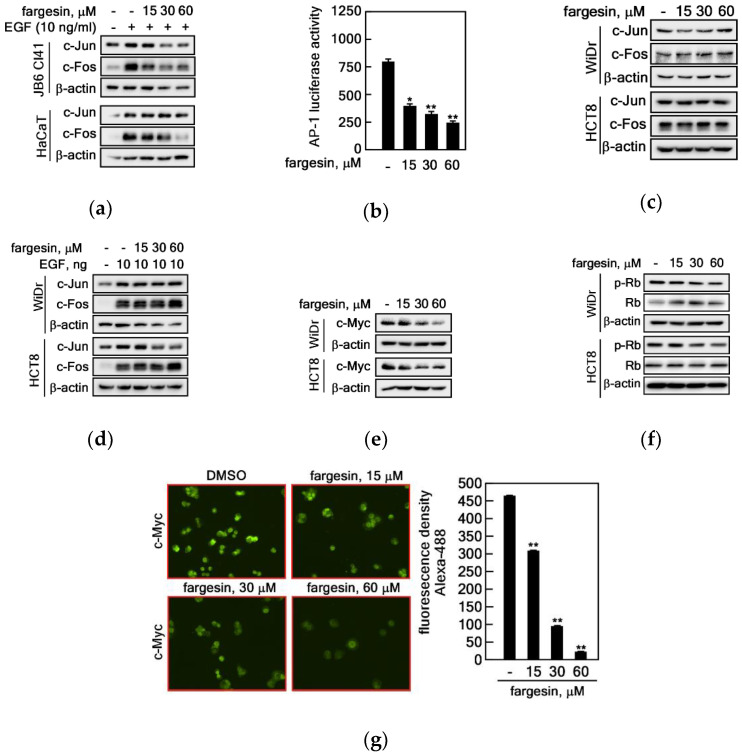
Differential regulation of AP-1 signaling and c-Myc-mediated CDK2/cyclin E signaling mediates cell proliferation differentially in premalignant and cancer cells. (**a**) Fargesin suppresses EGF-induced c-Jun and c-Fos protein levels in JB6 Cl41 and HaCaT cells. Cell lysates (30 μg) extracted from JB6 Cl41 and HaCaT cells were utilized to examine the protein levels of c-Jun and c-Fos by Western blotting using specific antibodies as indicated. (**b**) Fargesin inhibits AP-1 transactivation activity. JB6 Cl41 (5 × 10^4^) cells stably transfected with AP-1 luciferase reporter plasmid were subjected to AP-1 transactivation activity analysis as described in “Materials and Methods”. Data: a triplicate experiment; values: ±SEM; significance: * *p* < 0.05, ** *p* < 0.01 vs. non-fargesin-treated control by Student’s *t*-test. (**c**,**d**) Fargesin did not alter c-Jun and c-Fos protein levels in both normal and EGF-stimulated cell culture conditions in colon cancer cells. Cell lysates (30 μg) extracted from WiDr and HCT8 cells (**c**) and EGF-stimulated WiDr and HCT8 cells (**d**) were utilized to examine the protein levels of c-Jun and c-Fos by Western blotting using specific antibodies as indicated. (**e**) Fargesin suppresses c-Myc protein levels in colon cancer cells. Cell lysates (30 μg) extracted from WiDr and HCT8 cells were subjected to analyze c-Myc protein levels by Western blotting using the c-Myc specific antibody as indicated. (**f**) Fargesin suppresses phosphorylation of Rb protein. Cell lysates (30 μg) extracted from WiDr and HCT8 cells were subjected to analysis of phosphorylation levels of Rb protein levels by Western blotting using the phospho- and total-specific Rb antibodies as indicated. (**g**) Fargesin suppresses c-Myc protein levels in both the cytosol and nucleus. Right panels: WiDr (2 × 10^4^) cells were seeded into four-chamber slides, cultured overnight, and treated with the indicated doses of fargesin for 6 h. The cells were fixed, permeabilized, and hybridized with c-Myc-specific primary antibody and Alexa 488-conjugated secondary antibody. The c-Myc was observed under confocal microscope (×200). The original images with c-Myc (Alexa 488 shown as green) and DAPI (shown as blue) and the merged images are in Appendix A. Graphs: The fluorescence intensity was measured by the Image J computer program and normalized with DAPI intensity. Representative photographs were obtained from a photograph taken randomly at four different locations/chamber in three independent experiments. DAPI indicates nuclei. Data: a triplicate experiment; values: ±SEM; significance: ** *p* < 0.01 vs. non-fargesin-treated control by Student’s *t*-test. (**a**–**f**) β-actin was used to verify equal protein loadings.

## Data Availability

The data presented in this study are available in Appendix A here.

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
