# Peer review of "Fargesin Inhibits EGF-Induced Cell Transformation and Colon Cancer Cell Growth by Suppression of CDK2/Cyclin E Signaling Pathway"

_ijms, 2021, doi:10.3390/ijms22042073_

Round 1

Reviewer 1 Report

I have already received this email and have already double-checked the manuscript and given my approval for publication

Author Response

Feb-05-2021

Editor

RE: ijms-1082059-R1

(Fargesin-mediated suppression of CDK2/cyclin E signaling inhibited EGF-induced cell transformation and cancer growth of colon cancer cells)

Dear Editor:

With this letter, we submit our revision manuscript (ijms-1082059-R1) entitled “Fargesin inhibits EGF-induced cell transformation and colon cancer cell growth by suppression of CDK2/cyclin E signaling pathway” by Lee et al. for your consideration as a publication in the International Journal of Molecular Sciences. The work has not been previously published and is not under consideration for publication elsewhere. The authors have no conflict of interest that might be construed to have influenced the results or contents of the manuscript.

We added the results and discussed new data, point-by-point, in the revised manuscript. Thus, we believe the conclusions made in the revised manuscript are strongly supported by more robust results and discussions.

Reviewer 1

Comment: I have already received this email and have already double-checked the manuscript and given my approval for publication

Response: We specially thank the reviewer’s effort to improve our manuscript for the publication in IJMS.

Reviewer 2

Overall comment:

Comment 1. Title should be reframed and its making confusion for reader.

Response 1. We thank the reviewer for invaluable comment. We reframed the title of our manuscript as “Fargesin inhibits EGF-induced cell transformation and colon cancer cell growth by suppression of CDK2/cyclin E signaling pathway“.

Comment 2. Over all the study was well designed and executed. However, author should clarify following doubts before publication.

Response 2. We thank the reviewer for recognition. We added the explanation point-by-point in the revised manuscript (Please see highlighted area in the revised manuscript).

Specific comments:

Comment 1. Normally cell cycle analysis G0/G1 (40-60%), S1 (10-30%) and G2/M (10-30%), however Figure 1B, S phase higher than G0/G1 phase, which is not possible? Can you explain? Figure 1 C looks better but Fig 1B and 1C are contradict. 

Response 1. We thank the reviewer comment. We agreed with the reviewer’s opinion, because we also published a data with similar cell cycle population using HEK293T cells (Cancer Research 65(9):3596-3603, 2005). Based on our experience and other literatures, cell cycle population in normal cell culture condition is very vary depending on the cell types and cell lines. With steadiness, JB6 Cl41 cells, premalignant skin epidermal cells, showed about 40% G1/G0, about 55% S and 10% G2/M of cell cycle population (Cancer Research 67(17):8104-8112, 2007; Carcinogenesis 35(2):432-441, 2014; Molecular Carcinogenesis 58:1221-1233, 2019). We added citation at the “Results” section of the revised manuscript.

With our apology and thanks for not detail description in Figure 1 legend, the difference between Figure 1B and 1C was depended on the cell culture condition. Figure 1C is to evaluate the cell cycle transition from G1 to S, we starved cells for 24 h and then stimulated with EGF and/or indicated doses of fargesin as designed. Since starvation induces G1 cell cycle phase, the difference between Figure 1B and 1C is reasonable and acceptable. Thus, we corrected the Figure 1 legend and added a sentence with citations as “The cell cycle distribution of JB6 Cl41 and HaCaT cells in normal cell culture condition was similar to previous literatures” at “Results” section in the revised manuscript.

Comment 2. Figure 1 C showed clear G0/G1 arrest, are you sure there is no cell death? what about SubG1 %? Could you show cell cycle histogram data?

Response 2. We thank the reviewer comments. We had attached supplementary results in which contained the cytotoxicity of fargesin. In the revised manuscript also contained the cytotoxicity data and histogram of flowcytometry in supplementary Figure S2 and S3.

Comment 3. Again Figure 2 B S phase is higher than G0/G1.

Response 3. Please see response 1.

Comment 4. Why there is no p53 data since the fargesin mainly playing cell cycle relate inhibition and also increase p21?

Response 4. We thank the reviewer’ constructive comment. Since fargesin induced p21 protein levels and reduced cell cycle progress, fargesin-mediated p53 phosphorylation is questionable. I believe that our thinking is nearly similar with the reviewer. We found that fargesin induced phosphorylation of p53 at Ser37 in cancer cells. This work, the anticancer mechanisms of fargesin on the activation of p53 phosphorylation, is currently on going. Thus, I believe that the reviewer can understand our situation. Instead of the addition of p53 results, we discussed the issues at the “Discussion” section in the revised manuscript.

Comment 5. Introduction and discussion saying same information and too long introduction. 

Response 5. We thank the reviewer for constructive comment. We reduced the “Introduction” section in the revised manuscript. Moreover, we improved “Discussion” section in the revised manuscript.

Comment 6. Discussion should be improved rather than giving background.  

Response 6. Please see response 5.

Thus, we hope that you will agree that we have addressed each reviewer’s comments adequately and that this manuscript can now be considered for publication in International Journal of Molecular Sciences.

Sincerely,

Yong-Yeon Cho, Ph. D., Professor

College of Pharmacy, The Catholic University of Korea

43, Jibong-ro, Wonmi-gu, Bucheon-si, Gyeonggi-do 420-743, Republic of Korea

Phone: +82-2-2164-4092

Fax: +82-2-2164-4059

E-mail: yongyeon@catholic.ac.kr, choyycho@gmail.com

Reviewer 2 Report

Title should be reframed and its making confusion for reader. Over all the study was well designed and executed however author should clarify following doubts before publication.

  1. Normally cell cycle analysis G0/G1 (40-60%), S1 (10-30%) and G2/M (10-30%), however Figure 1B, S phase higher than G0/G1 phase, which is not possible? Can you explain? Figure 1 C looks better but Fig 1B and 1C are contradict. 
  2. Figure 1 C showed clear G0/G1 arrest, are you sure there is no cell death? what about SubG1 %? Could you show cell cycle histogram data?
  3. Again Figure 2 B S phase is higher than G0/G1.
  4. Why there is no p53 data since the  fargesin mainly playing cell cycle relate inhibition and also increase p21?
  5. Introduction and discussion saying same information and too long introduction. 
  6. Discussion should be improved rather than giving background.  

Author Response

Feb-05-2021

Editor

RE: ijms-1082059-R1

(Fargesin-mediated suppression of CDK2/cyclin E signaling inhibited EGF-induced cell transformation and cancer growth of colon cancer cells)

Dear Editor:

With this letter, we submit our revision manuscript (ijms-1082059-R1) entitled “Fargesin inhibits EGF-induced cell transformation and colon cancer cell growth by suppression of CDK2/cyclin E signaling pathway” by Lee et al. for your consideration as a publication in the International Journal of Molecular Sciences. The work has not been previously published and is not under consideration for publication elsewhere. The authors have no conflict of interest that might be construed to have influenced the results or contents of the manuscript.

We added the results and discussed new data, point-by-point, in the revised manuscript. Thus, we believe the conclusions made in the revised manuscript are strongly supported by more robust results and discussions.

Reviewer 1

Comment: I have already received this email and have already double-checked the manuscript and given my approval for publication

Response: We specially thank the reviewer’s effort to improve our manuscript for the publication in IJMS.

Reviewer 2

Overall comment:

Comment 1. Title should be reframed and its making confusion for reader.

Response 1. We thank the reviewer for invaluable comment. We reframed the title of our manuscript as “Fargesin inhibits EGF-induced cell transformation and colon cancer cell growth by suppression of CDK2/cyclin E signaling pathway“.

Comment 2. Over all the study was well designed and executed. However, author should clarify following doubts before publication.

Response 2. We thank the reviewer for recognition. We added the explanation point-by-point in the revised manuscript (Please see highlighted area in the revised manuscript).

Specific comments:

Comment 1. Normally cell cycle analysis G0/G1 (40-60%), S1 (10-30%) and G2/M (10-30%), however Figure 1B, S phase higher than G0/G1 phase, which is not possible? Can you explain? Figure 1 C looks better but Fig 1B and 1C are contradict. 

Response 1. We thank the reviewer comment. We agreed with the reviewer’s opinion, because we also published a data with similar cell cycle population using HEK293T cells (Cancer Research 65(9):3596-3603, 2005). Based on our experience and other literatures, cell cycle population in normal cell culture condition is very vary depending on the cell types and cell lines. With steadiness, JB6 Cl41 cells, premalignant skin epidermal cells, showed about 40% G1/G0, about 55% S and 10% G2/M of cell cycle population (Cancer Research 67(17):8104-8112, 2007; Carcinogenesis 35(2):432-441, 2014; Molecular Carcinogenesis 58:1221-1233, 2019). We added citation at the “Results” section of the revised manuscript.

With our apology and thanks for not detail description in Figure 1 legend, the difference between Figure 1B and 1C was depended on the cell culture condition. Figure 1C is to evaluate the cell cycle transition from G1 to S, we starved cells for 24 h and then stimulated with EGF and/or indicated doses of fargesin as designed. Since starvation induces G1 cell cycle phase, the difference between Figure 1B and 1C is reasonable and acceptable. Thus, we corrected the Figure 1 legend and added a sentence with citations as “The cell cycle distribution of JB6 Cl41 and HaCaT cells in normal cell culture condition was similar to previous literatures” at “Results” section in the revised manuscript.

Comment 2. Figure 1 C showed clear G0/G1 arrest, are you sure there is no cell death? what about SubG1 %? Could you show cell cycle histogram data?

Response 2. We thank the reviewer comments. We had attached supplementary results in which contained the cytotoxicity of fargesin. In the revised manuscript also contained the cytotoxicity data and histogram of flowcytometry in supplementary Figure S2 and S3.

Comment 3. Again Figure 2 B S phase is higher than G0/G1.

Response 3. Please see response 1.

Comment 4. Why there is no p53 data since the fargesin mainly playing cell cycle relate inhibition and also increase p21?

Response 4. We thank the reviewer’ constructive comment. Since fargesin induced p21 protein levels and reduced cell cycle progress, fargesin-mediated p53 phosphorylation is questionable. I believe that our thinking is nearly similar with the reviewer. We found that fargesin induced phosphorylation of p53 at Ser37 in cancer cells. This work, the anticancer mechanisms of fargesin on the activation of p53 phosphorylation, is currently on going. Thus, I believe that the reviewer can understand our situation. Instead of the addition of p53 results, we discussed the issues at the “Discussion” section in the revised manuscript.

Comment 5. Introduction and discussion saying same information and too long introduction. 

Response 5. We thank the reviewer for constructive comment. We reduced the “Introduction” section in the revised manuscript. Moreover, we improved “Discussion” section in the revised manuscript.

Comment 6. Discussion should be improved rather than giving background.  

Response 6. Please see response 5.

Thus, we hope that you will agree that we have addressed each reviewer’s comments adequately and that this manuscript can now be considered for publication in International Journal of Molecular Sciences.

Sincerely,

Yong-Yeon Cho, Ph. D., Professor

College of Pharmacy, The Catholic University of Korea

43, Jibong-ro, Wonmi-gu, Bucheon-si, Gyeonggi-do 420-743, Republic of Korea

Phone: +82-2-2164-4092

Fax: +82-2-2164-4059

E-mail: yongyeon@catholic.ac.kr, choyycho@gmail.com

This manuscript is a resubmission of an earlier submission. The following is a list of the peer review reports and author responses from that submission.

Round 1

Reviewer 1 Report

In this manuscript the authors describe the results of a research on fargesin, a lignan major ingredient in Shin-Yi. In this study, they observed that fargesin inhibited cell proliferation and transformation by suppression of EGF-stimulated G1/S-phase cell cycle transition in premalignant JB6 Cl41 and HaCaT cells.

The paper is well written from all points of view and deserves to be published in this journal.

line 219 was further was increased by fargesin: too much "was"

line 221-223 By analyzing the PKB signaling pathway, we observed that EGF induced phosphorylation of mTOR at Ser2448, and AKT at Thr308 and Ser473.

did you find these results in a previous search? you can put a bibliographic reference

line 386-388 . These results demonstrated that slight differences in chemical structure might produce large differences in molecular targeting of proteins that are involved in human diseases including cancers.

At this point one could make very short activity structure reports. They are all very similar molecules but you have data on the biological activity of many of the lignans contained in Shin-Yi   428-430 . The results of this study strongly ndicate that fargesin has potential as a chemoprevention and/or therapeutic agent against human cancer. It would also be appropriate to include a data on sellectivity in cytotoxicity, eg. on fibroblasts.

Author Response

Reviewer 1.

Comment 1. In this manuscript the authors describe the results of a research on fargesin, a lignan major ingredient in Shin-Yi. In this study, they observed that fargesin inhibited cell proliferation and transformation by suppression of EGF-stimulated G1/S-phase cell cycle transition in premalignant JB6 Cl41 and HaCaT cells. The paper is well written from all points of view and deserves to be published in this journal.

Response 1. We thank the reviewer for the recognition of our research. Your encouraging comments are very much appreciated.

Comment 2. line 219 was further was increased by fargesin: too much "was".

Response 2. We apologize for our carelessness. We changed the sentence to “Interestingly, unexpected results repeatedly showed that EGF-induced ERK and RSK’s phosphorylation was more increased by fargesin treatment in a dose-dependent manner (new Figure 3b)” in the revised manuscript.

Comment 3. line 221-223 By analyzing the PKB signaling pathway, we observed that EGF induced phosphorylation of mTOR at Ser2448, and AKT at Thr308 and Ser473. Did you find these results in a previous search? You can put a bibliographic reference

Response 3. We thank the reviewer’s comments. Our previous research has showed the increase of phospho-mTOR at Ser2448(in Fig. 2B in Carcinogenesis 36(10): 1223–1234, 2015), and phospho-AKT at Thr308 and Ser473 by EGF stimulation (in Fig. 2B in Carcinogenesis 35(2):432-441, 2014; in Fig. 2B in Carcinogenesis 36(10):1223–1234, 2015). Thus, we re-wrote the sentence with reference citations as “By analyzing the PKB signaling pathway, EGF-induced mTOR phosphorylation at Ser2448, and AKT at Thr308 and Ser473 was observed (Figure 2c) as shown as our previous reports [11, 12]” in the revised manuscript.

Comment 4. line 386-388. These results demonstrated that slight differences in chemical structure might produce large differences in molecular targeting of proteins that are involved in human diseases including cancers. At this point one could make very short activity structure reports. They are all very similar molecules but you have data on the biological activity of many of the lignans contained in Shin-Yi.

Response 4. We thank the reviewer for an invaluable point. We agreed with the reviewer’s opinion for the activity regulation of lignan compounds, which contain very similar structures, but quiet different activity and target molecules. Currently, we are preparing a review manuscript with a viewpoint of medicinal chemistry. To help the reader’s understanding, we described the molecular targets of magnolin, aschantin, and epimagnolin with key interactions with each target protein in the “Discussion” section.

Comment 5. In lane 428-430. The results of this study strongly indicate that fargesin has potential as a chemoprevention and/or therapeutic agent against human cancer. It would also be appropriate to include a data on selectivity in cytotoxicity, eg. on fibroblasts.

Response 5. We thank the reviewer’s invaluable comments. Unfortunately, we could not find the target(s) of fargesin. Thus, selectivity for fargesin cannot be measured in our current study. Instead of the selectivity, the toxicity data measured in JB6 Cl41 cells has added in the revised manuscript (Supplemental Figure S2). The results indicated that fargesin did not show cytotoxicity over 60 µM in cell culture condition.

Reviewer 2 Report

In the manuscript “Fargesin-mediated suppression of CDK2/cyclin E signaling inhibited EGF-induced cell transformation and cancer growth of colon cancer cells” by Lee et al., the authors use the lignan compound fargesin in in vitro culture conditions and tested its influence on cell growth and differentiation of some tumor cell lines, such as JB6 Cl41 and HaCaT cells as well as HCT116, WiDr and HCT8 cells.

The authors find that fargesin suppressed cell proliferation in a dose-dependent manner. Furthermore, fargesin induced morphological changes in some colon cancer cell lines.

In some cells, the authors found that phosphorylation of components of the MAPK signalling pathway (i.e. ERKs and RSKs) was increased by fargesin, while phosphorylation of components of the PI3K / PKB signalling pathway (i.e. mTOR and AKT) was diminished by fargesin. However, in other cell lines the authors found that diminished cell proliferation and colony growth was associated with the modulation of cell cycle regulators via CDK2/cyclin E/p21WAF1/Cip1 complex formation.

The data are pure descriptive and there is no speculation on the molecular basis of the observed results. I would not see how such a study could be published anywhere.

Author Response

Reviewer 2.

Comments. In the manuscript “Fargesin-mediated suppression of CDK2/cyclin E signaling inhibited EGF-induced cell transformation and cancer growth of colon cancer cells” by Lee et al., the authors use the lignan compound fargesin in in vitro culture conditions and tested its influence on cell growth and differentiation of some tumor cell lines, such as JB6 Cl41 and HaCaT cells as well as HCT116, WiDr and HCT8 cells. The authors find that fargesin suppressed cell proliferation in a dose-dependent manner. Furthermore, fargesin induced morphological changes in some colon cancer cell lines. In some cells, the authors found that phosphorylation of components of the MAPK signaling pathway (i.e. ERKs and RSKs) was increased by fargesin, while phosphorylation of components of the PI3K / PKB signaling pathway (i.e. mTOR and AKT) was diminished by fargesin. However, in other cell lines the authors found that diminished cell proliferation and colony growth was associated with the modulation of cell cycle regulators via CDK2/cyclin E/p21WAF1/Cip1 complex formation. The data are pure descriptive and there is no speculation on the molecular basis of the observed results. I would not see how such a study could be published anywhere.

Response. We thank the reviewer for the recognition of our research. We will continue to decipher the molecular target(s) of fargesin.

Reviewer 3 Report

The manuscript of Lee and colleagues deals with the antiproliferative effect of fargesine. This is an observational and descriptive study. The main problems are: a) in some cases, the data reported in the figures do not support the claims in the text; b) most of the effect of fargesine are only associated each to another, there is no demonstration of a linkage and this should be considered in the discussion.

Specific points:

1) A complete check out and revision of language is needed

2) Line 115.  “Both JB6 Cl41 and HaCaT cells showed 50% inhibitory effect of cell proliferation at, approximately, 30 µM fargesin concentration (Figure 1A)”  The data in Figure 1A do not support this sentence. The figure shows a significantly higher effect  of fargesine in JB6 Cl41 cells.  The exact calculated values of I50 should be reported. Please, also control the y axe of fig. 1A.

3) Line 170. Again, data depicted in fig. 2A are not consistent with the text.The calculated values of I50 for each cell line should be reported.

4) Fig 2A. Statistics is reported only in panel HTC8. The p description is different in the panel and in the figure caption (last line).

5) Line 224. Fig 3C shows that AKT phosphorylation (T308) is not decreased, but actually increased by fargesine.

6) Line 228. Fig 3D does not support the claim of a suppression of phosphorylation of ERKs and RSKs in WiDr cells. Further, the panel showing p-RSK in HCT8 cells has a very low graphic quality.

7) Line 229. Figure 3E does not depict any effect of fargesine on mTOR phosphorylation in any cell line.

8) Lines 230-233. The sentence should be deleted (even in the discussion section). There is no demonstration that the effect of fargesine on cell shape is a consequence of the (quite limited) effect on the phosphorylation of signaling proteins.

9) Line 225. “were regulated by” should be changed in “were associated to”.

10) Fig. 4a. The panel is the same as fig. 2B.

11) Lines 260-262. Again, the sentence is not fully consistent with data in fig. 4B.

12) Line 271. “are evidence” should be changed in “suggest”.

13) Lines 310-311. Fargesine does not decrease c-Jun level in HaCaT cells according to fig. 5A.

Author Response

Reviewer 3.

General Comments. The manuscript of Lee and colleagues deals with the antiproliferative effect of fargesin. This is an observational and descriptive study. The main problems are: a) in some cases, the data reported in the figures do not support the claims in the text; b) most of the effect of fargesin are only associated each to another, there is no demonstration of a linkage and this should be considered in the discussion.

Response for general comments: We thank the reviewer for constructive and invaluable comments. We carefully read and corrected the manuscript in an accordance with reviewer’s suggestions. We changed our manuscript to dissolve the two main points: a) the manuscript should have clarity to support our findings with the scientific sound, and b) careful discussion with linkages based on our finding with scientific logic. We described point-by-point in an accordance with the reviewer’s suggestions and highlighted in the revised manuscript.

Specific points:

Comment 1. A complete check out and revision of language is needed

Response 1. We apologize for our carelessness. We have carefully checked our whole manuscript and corrected the revised manuscript. The changed sentences and areas are highlighted.

Comment 2. Line 115. “Both JB6 Cl41 and HaCaT cells showed 50% inhibitory effect of cell proliferation at, approximately, 30 µM fargesin concentration (Figure 1A)”. The data in Figure 1A do not support this sentence. The figure shows a significantly higher effect of fargesine in JB6 Cl41 cells.  The exact calculated values of IC50 should be reported. Please, also control the y axe of fig. 1A.

Response 2. We thank the reviewer comments. We agreed with the reviewer’s opinion. We corrected the sentence with more accurate information in the revised manuscript. We also re-checked our all Figures for mistypes in the revised manuscript.

Comment 3. Line 170. Again, data depicted in fig. 2A are not consistent with the text. The calculated values of IC50 for each cell line should be reported.

Response 3. We apologize to the reviewer for our inattention. We added calculated IC50 for each cell in the revised manuscript.

Comment 4. Fig 2A. Statistics is reported only in panel HTC8. The p description is different in the panel and in the figure caption (last line).

Response 4. We apologize for our carelessness. We have checked all statistical data and then confirmed. Moreover, we re-checked Figure legends for the correct description in the revised manuscript.

Comment 5.  Line 224. Fig 3C shows that AKT phosphorylation (T308) is not decreased, but actually increased by fargesin.

Response 5. We thank the reviewer comment. We corrected the sentences with accurate translation as “By analyzing the PKB signaling pathway, EGF-induced mTOR phosphorylation at Ser2448, and AKT at Thr308 and Ser473 was observed (Figure 3c) as shown as our previous reports [11, 12]. In contrast to ERKs and RSKs, phosphorylation of mTOR at Ser2448, and AKT at Thr308, and Ser473 was decreased by fargesin treatment in a dose-dependent manner (Figure 3c). However, we did not know the reasons why AKT phosphorylation at Thr308 by co-treatment of EGF and 15 μM of fargesin was increased (Figure 3c).” in the Results section of the revised manuscript.

Comment 6. Line 228. Fig 3D does not support the claim of a suppression of phosphorylation of ERKs and RSKs in WiDr cells. Further, the panel showing p-RSK in HCT8 cells has a very low graphic quality.

Response 6. We thank the reviewer’s constructive comments. We agreed with the reviewer’s comments. Thus, we have changed with a new picture in the revised manuscript.

Comment 7. Line 229. Figure 3E does not depict any effect of fargesin on mTOR phosphorylation in any cell line.

Response 7. We thank the reviewer comments. We replaced the Western blot panels for phosphorylation of mTOR-S2448 in the revised manuscript.

Comment 8. Lines 230-233. The sentence should be deleted (even in the discussion section). There is no demonstration that the effect of fargesin on cell shape is a consequence of the (quite limited) effect on the phosphorylation of signaling proteins.

Comment 8. We thank the reviewer’s invaluable comments. We agreed with the reviewer’s opinion. Thus, we changed the sentence as in “These results indicated that fargesin-mediated inhibition of ERKs, and PKB signaling pathways suppresses colon cancer cell proliferation and colony growth” in the revised manuscript.

Comment 9. Line 225. “were regulated by” should be changed in “were associated to”.

Response 9. The reviewer point “lane 225” may be lane 255. We agreed with the reviewer’s opinion. Thus, we have changed in “were associated to” in the revised manuscript.

Comment 10. Fig. 4a. The panel is the same as fig. 2B.

Response 10. We apologize for our carelessness. We deleted old Figure 4A. The citation for old “Figure 4a” has been changed to “Figure 2b” in the revised manuscript. Moreover, we made the changes for figure citations for Figure 4 in the Results section and Figure legends for Figure 4 in the revised manuscript.

Comment 11. Lines 260-262. Again, the sentence is not fully consistent with data in fig. 4B.

Response 11. We apologize to the reviewer for our inattention. We described the data with detail and accuracy for the new Figure 4a in the revised manuscript.

Comment 12. Line 271. “are evidence” should be changed in “suggest”.

Response 12. We thank the reviewer comments. We agreed with the reviewer’s opinion. We have changed to “suggest” in the revised manuscript.

Comment 13. Lines 310-311. Fargesin does not decrease c-Jun level in HaCaT cells according to fig. 5A.

Response 13. We apologize for mistranslation. We have corrected the sentence to “We found that EGF stimulation induced c-Jun and c-Fos levels in JB6 Cl41 cells (Figure 5a). Notably, JB6 Cl41 cells showed the decrease of c-Jun and c-Fos protein levels by fargesin treatment in a dose-dependent manner, whereas HaCaT cells showed the decrease of only c-Fos, but not in c-Jun, protein levels by fargesin in a dose-dependent manner (Figure 5a).” in the revised manuscript.

Round 2

Reviewer 1 Report

the manuscript has been carefully and painstakingly corrected and is ready for publication

Reviewer 3 Report

Thanks to authors for they corrections that have improved the quality of the manuscript.